# The Management of Intracranial Aneurysms: Current Trends and Future Directions

**Aviraj S. Deshmukh** [1], **Stefano M. Priola** [2,*], **Aris H. Katsanos** [3], **Gianluca Scalia** [4], **Aderaldo Costa Alves** [2], **Abhilekh Srivastava** [3] and **Christine Hawkes** [5]

1    Division of Clinical Sciences, Health Sciences North, Northern Ontario School of Medicine University, Sudbury, ON P3E 2C6, Canada; avirajdeshmukh@gmail.com
2    Division of Neurosurgery, Health Sciences North, Northern Ontario School of Medicine University, Sudbury, ON P3E 2C6, Canada; acostaalves@nosm.ca
3    Division of Neurology, Hamilton General Hospital, McMaster University, Hamilton, ON L8S 4L8, Canada
4    Department of Neurosurgery, Highly Specialized Hospital of National Importance "Garibaldi", 95126 Catania, Italy; gianluca.scalia@outlook.it
5    Division of Neurology, Department of Medicine, Sunnybrook Health Sciences Centre, University of Toronto, Toronto, ON M5S 1A1, Canada; christine.hawkes@sunnybrook.ca
*    Correspondence: spriola@hsnsudbury.ca; Tel.: +1-(705)-806-2248; Fax: +1-(705)-806-2998

**Abstract:** Intracranial aneurysms represent a major global health burden. Rupture of an intracranial aneurysm is a catastrophic event. Without access to treatment, the fatality rate is 50% in the first 30 days. Over the last three decades, treatment approaches for intracranial aneurysms have changed dramatically. There have been improvements in the medical management of aneurysmal subarachnoid haemorrhage, and there has been an evolution of treatment strategies. Endovascular therapy is now the mainstay of the treatment of ruptured intracranial aneurysms based on robust randomised controlled trial data. There is now an expansion of treatment indications for unruptured intracranial aneurysms to prevent rupture with both microsurgical clipping and endovascular treatment. Both microsurgical and endovascular treatment modalities have evolved, in particular with the introduction of innovative endovascular treatment options including flow diversion and intra-saccular flow disruption. These novel therapies allow clinicians to treat more complex and previously untreatable aneurysms. We aim to review the evolution of treatment strategies for intracranial aneurysms over time, and discuss emerging technologies that could further improve treatment safety and functional outcomes for patients with an intracranial aneurysm.

**Keywords:** endovascular coiling; surgical clipping; unruptured intracranial aneurysm; flow diverter; flow disruptors





## 1. Introduction

Intracranial aneurysms (IA) are focal pathological dilatations of the cerebral arteries. While typically asymptomatic, the rupture of an IA leading to SAH is responsible for 5% of all strokes [1]. IA account for 80–85% of non-traumatic subarachnoid haemorrhage (SAH), but can also lead to intraparenchymal or subdural haemorrhage [1]. The average incidence of aneurysmal bleeding is approximately 9 per 100,000 person-years with the highest rates reported in Japanese and Finnish populations [2]. Though the annual incidence rates of ruptured IA are relatively stable, unruptured intracranial aneurysms (UIA), are increasingly detected in clinical practice, due to the widespread availability of advanced cross-sectional neuroimaging. The prevalence of IA is variable and depends on the evaluation methods used but IA are estimated to be found in 2–3.2% of the general population with a male to female ratio of 1:2 [3].

Aneurysmal subarachnoid haemorrhage is a catastrophic event with very high morbidity and mortality. About 15% of patients with ruptured IA could not even reach a hospital

and there is a 50% thirty-day mortality in the remaining survivors [4]. Compared to other subtypes of stroke, patients affected due to SAH tend to be younger, which results in a greater loss of productive life [5]. Rebleeding is the most imminent danger; which drastically decreases the chances of a good outcome. The risk of rebleeding is at its maximum in the initial 24 h, during which the risk ranges from 4.1% to 17.3% [6]. Therefore, early diagnosis and aneurysm occlusion is a priority. Nowadays, many centres tend to operate within the first 24 h of onset via an endovascular or open surgical modality. On the other hand, unruptured aneurysms, with an annual risk of rupture 0.49–1.8%, represent a challenge in terms of conservative management versus aggressive intervention [7]. Further, selecting an appropriate treatment strategy is a complex decision, which varies according to patient and aneurysm-related factors. The aim of this article is to provide a comprehensive review of the evolution of these two treatment modalities over the years as well as to provide a valuable and relevant supplement to existing knowledge in terms of newer devices and techniques of aneurysm management.

## 2. Aneurysm Pathophysiology

The cerebrovascular system is inherently susceptible to aneurysm formation, although the aetiology of these abnormalities may be diverse. The important ones include the absence of an external elastic lamina, paucity of supportive perivascular tissues, and attenuated tunica media in the intracranial arterial system [8]. The distribution of cerebral aneurysms around the bifurcations or branch location points towards hemodynamic factors and/or wall shear stress as a principal determinant for aneurysm formation, growth and rupture (Figure 1A). Incidence of IA is high in females, smokers, hypertensives, and a few geographical regions. Apart from this, multiple familial conditions are associated with a high incidence of IA, such as autosomal dominant inherited polycystic kidney disease, Marfan syndrome, fibromuscular dysplasia (FMD), alpha1-antitrypsin deficiency, etc. (Table 1). The list is exhaustive and illustrative of the fact that multiple genetic, hemodynamic and environmental factors act together in the formation and growth of IA [8].

**Table 1.** Risk factors for aneurysm formation.

| |
| --- |
| 1. Female Sex |
| 2. Smoking |
| 3. Hypertension |
| 4. Coarctation of the aorta |
| 5. Racial Predisposition e.g., Japanese, Finnish populations |
| 6. Hereditary syndromes e.g., Ehlers-Danlos syndrome, Pseudoxanthoma elasticum, Polycystic Kidney Disease, FMD |
| 7. Familial aneurysm |

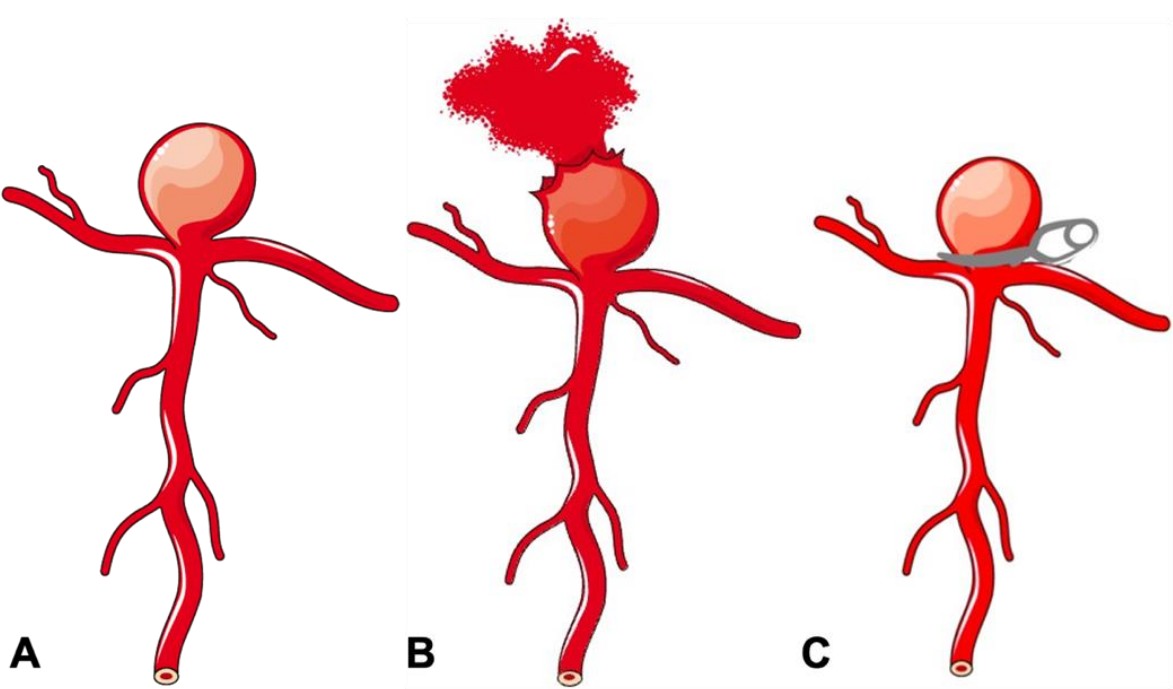

**Figure 1.** Schematic diagram of a bifurcation aneurysm. (**A**) Bifurcation aneurysm. (**B**) Ruptured bifurcation aneurysm. (**C**) Surgical clipping of bifurcation aneurysm.

### 3. Aneurysm Classification

IA can be classified using several schemes (Table 2). Clinically, they can be classified on the basis of rupture status: ruptured or unruptured. According to morphology, aneurysms are classified into saccular and non-saccular types. Non-saccular IA are further classified into fusiform, dolichoectatic, and dissecting aneurysms. According to location in the intracranial circulation, aneurysms are divided into anterior (90%) and posterior circulation (10%) aneurysms. IA are also classified by size into small (<10 mm), large (10–25 mm), and giant (>25 mm) categories. Aneurysm angioarchitecture is important for planning suitability for endovascular management and can thus be classified according to neck size and relationship with the dome. Apart from serving a descriptive purpose, these classification systems also help in predicting prognosis, plan management, and appropriate treatment strategies. For example, despite advances in knowledge about aneurysm pathophysiology and technology, aneurysms with a large size (>10 mm), wide neck, unfavourable dome-to-neck ratio (<2), posterior circulation location and fusiform configuration remain therapeutic challenges with >20% faring poorly despite the best endovascular or surgical treatment [9,10]. Our review is focussed on the management of the saccular aneurysm, which is the single largest group of IA (Figure 1A).

**Table 2.** Aneurysm Classification.

| Classification | Description |
|---|---|
| Rupture Status | Unruptured, Ruptured |
| Shape | Saccular, Fusiform, Dissecting |
| Location | Anterior Circulation (Anterior Communicating Artery, Middle Cerebral Artery aneurysm, etc.), Posterior Circulation (Basilar Artery, Vertebral artery aneurysm, etc.) |
| Size | Small (<10 mm), Large (>10–25 mm), Giant (>25 mm) |
| Aetiology | Atherosclerotic, Mycotic, Traumatic, Congenital, Hemodynamic (Flow-related), Idiopathic |

## 4. The Evolution of and Current Surgical Treatment Options

Though first clip surgery was performed in 1937, the origin of aneurysm surgery dates back to the early 19th century. In 1808, Dr. Cooper successfully treated the carotid aneurysm in the cervical region, based on the Hunterian principal of proximal artery ligation [11,12]. Despite high operative morbidity and mortality, proximal ligation technique remained the only option for aneurysm treatment for the next century. In 1911, Harvey Cushing first described the use of silver clips to occlude the vessels which were otherwise inaccessible to the ligature [13]. It was in 1937 that Dandy made use of these clips to perform neck ligation of saccular IA [14], which marks the era of the development of aneurysm clip surgery. The next major milestone in the evolution of aneurysm surgery was the development of operating microscopes in 1957. With the initial description of its use by Kurze et al., and later by Lougheed et al., microsurgery rapidly gained prominence through the discipline of neurosurgery and became a standard of care [15,16]. In the following decades, advancements in clip design and clip applicators emerged, facilitating improved visualization, preservation of adjacent arteries or nerves near the aneurysm neck, and enhancing surgical proficiency. The initial design of the clip did not allow reopening once placed and hence required absolute precision and accuracy. Norlen and Olivecrona overcame this limitation by adding winged blades to clips to allow reopening if placement was suboptimal [17]. Later modification involved the addition of serrations on blades to increase their purchase and to minimize the risk of slippage and aneurysm crushing [18]. Drake further refined clip design to facilitate convenient access to the aneurysm neck, leading to the development of contemporary fenestrated aneurysm clip designs [19]. Further modifications to the aneurysm clip were based on metallurgy to make them compatible with magnetic resonance imaging and to provide predictable closing pressures [20]. Changes in the clip applicator design allowed it to be slimmer, with a better clip–applicator relationship [17]. Some of the important milestones in the evolution of aneurysm treatment are summarized in Figure 2.

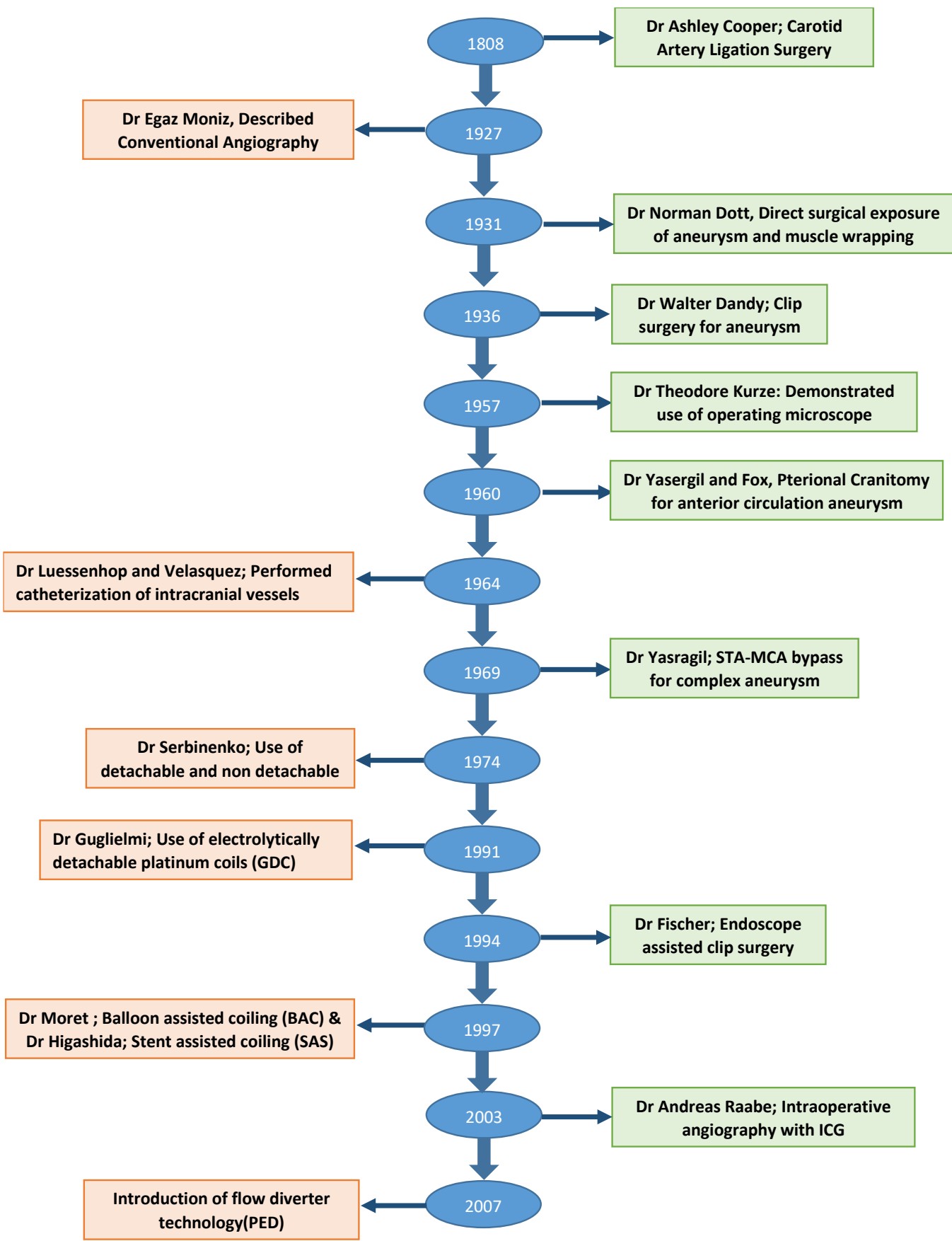

**Figure 2.** Flow diagram showing important milestones in the evolution of aneurysm treatment.

## 5. Clip Surgery

Clip surgery remained the effective and dominant treatment option for aneurysm surgery until the last decade of the 20th century. The main principles of clip surgery are to place a clip across the neck of the aneurysm to exclude the aneurysm from the circulation, while maintaining the patency of the parent vessel (Figure 1C). Therefore, the paramount aspect of clipping surgeries lies in good neck exposure. Over the years, the development of various skull-based approaches to aneurysm surgery helped to reduce the distance between surgeon and aneurysm, minimizing the exposure and retraction of neurovascular tissue and improving surgical manoeuvrability. Dandy himself described the frontolateral approach for aneurysm surgery [17]. However, the pterional approach (PTC, pterional craniotomy) described by Yasargil and Fox received widespread acceptance and became the standard of care [21]. PTC involved less brain retraction, but more bone removal and extensive retraction of temporal muscle. To address these limitations, several miniature versions have been created. These encompass supraorbital (SOC), lateral supraorbital (LSOC), mini-pterional (MPTC), and interhemispheric keyhole craniotomies [22–24]. Essentially, most of them are modelled on the standard PTC, with a customized opening location, trajectory, and angle of approach. The aim is to minimize unnecessary temporalis dissection and brain exposure, while maintaining relevant operative corridors.

A recent advancement happening in the area of the approach to aneurysms is the advent of endoscope in clip surgery. In 1994, Fischer et al. reported the first endoscope-assisted clip surgery [25]. The endoscope provides better in-depth illumination of surgical fields, extends viewing angles, and helps in the clear depiction of anatomy. Their aim is to complement standard microscopes, which are restricted to illumination and magnification along a line of sight. There are few case reports of treating aneurysms with only endoscopic transcranial and endonasal approaches [26]. Another recent advancement is the development of simple, rapid, and effective intraoperative angiography, employing fluorescent dyes. In 1994, Wrobel et al., described the use of fluorescein sodium angiography and in 2003, Andreas Raabe et al. introduced the use of intraoperative near-infrared indocyanine green (ICG) in video angiography [27,28]. Intraoperative angiography enables the assessment of aneurysm obliteration as well as the patency of the major vessels and small perforators, the important factors which determine technical success. The technique is less invasive than intraoperative digital subtraction angiography (DSA), but only vessels that can be seen by the operating microscope are evaluated. Another commonly used technique in aneurysm surgery is the use of intra-operative temporary artery occlusion (TAO), which shrinks the aneurysm and affords the operator better visibility and operating space. The occlusion time is normally limited to between 10 and 20 min and multiple occlusions can be applied with an intervening gap of 15 min for complex cases [29]. The use of intra-operative angiography and TAO has helped to greatly reduce the risk of intraoperative aneurysm rupture.

In the current era of endovascular therapy, an increasing number of aneurysms are undergoing treatment through this method. As a result, clip surgery is often reserved for complex cases. At the same time, the use of minimally invasive surgical approaches continues to evolve, focussing on safety and efficacy while maximizing patient comfort with cosmesis. The technique of neuroendoscopy may further reduce the invasiveness and use of operating microscopes from clip surgery.

## 6. Arterial Occlusion and Bypass Technique

The aim of bypass surgery is the isolation of aneurysmal lesions via the occlusion of the parent inflow artery and establishment of blood flow distally through the bypass. It was first reported by Crowell and Yasargil in 1969 for treating complex IA [30] and it is one the treatment options for giant IA, when direct clipping or endovascular repair is not possible. Bypasses can be extracranial (EC) to intracranial (IC) or intracranial (IC) to intracranial (IC) [31]. EC–IC bypass consists of two types: low flow and high flow bypass. In low flow bypass, the superficial temporal artery (STA) is anastomosed to an intracranial

artery. While in high-flow bypass, the common carotid artery (CCA) or external carotid artery (ECA) is anastomosed to an intracranial artery (CCA–IC or ECA–IC) using either the great saphenous vein (GSV) or radial artery (RA) as a conduit [32]. The IC–IC bypass involves removing the lesion and restoring the patency of the inflow and outflow arteries, with or without grafting, to establish a tension-free anastomosis [33]. IC–IC bypasses are less vulnerable to occlusion or injury and do not require a donor graft, however they are technically more demanding and can only be performed in limited locations where donor and recipient arteries lie in parallel and in close proximity [33].

## 7. Wrapping Technique

In this approach, the lesion is wrapped with autogenous tissue or absorbable material to reconstruct the integrity of the vessel wall. Though first described by Norman Dott in 1931, wrapping should never be the primary goal of surgery [34]. Rather, it is reserved for situations in which little else is possible. Wrapping can be performed with various materials such as cotton, cellulose fabric, muscle, or dura [35]. Wrapping and clipping was sometimes recommended for dissecting aneurysms where simple clipping is hazardous due to the friable wall of the aneurysm and parent artery [35].

## 8. The Evolution of Current Endovascular Treatment Options

While Egaz Moniz pioneered diagnostic cerebral angiography in 1927, the development of the endovascular approach to aneurysm treatment took several years to materialize [36]. In 1964, Luessenhop and Velasquez performed the first catheterization of intracranial vessels [37]. Following this, in 1974, the advent of the use of both detachable and non-detachable balloons was described by Serbinenko in his study involving 300 patients, marking the beginning of the balloon era in the endovascular treatment of intracranial aneurysms [38]. The rigid structure of the balloon produced an angioplasty of the aneurysm wall with consequent immediate or delayed aneurysm rupture. As a result, the use of balloons remained limited to the small clinical series of inoperable aneurysms. The initiation of the modern era of neuroendovascular therapy occurred with the development and utilization of electrolytically detachable platinum coils (GDC) in 1991 by Guglielmi et al. [39]. These coils were soft, retrievable, and detachable by the operator, which solved many of the problems with other earlier techniques. With continued advances in endovascular technology, aneurysm coiling quickly emerged as an accepted and viable alternative to surgical clipping over the last decade of the 20th century.

### 8.1. Aneurysm Coiling

Once approved by the US Food and Drug Administration (FDA) in 1995, aneurysm coiling became the primary treatment modality of IA in numerous centres [40,41]. The goal of coiling is to achieve dense packing and induce rapid coagulation within the aneurysm sac, thereby isolating it from active circulation. Aneurysm geometry is an important determinant of treatment decision and outcome. Unassisted coiling is typically appropriate for intracranial aneurysms (IA) with favourable anatomical characteristics such as neck width, dome-to-neck ratio, aspect ratio, and relationship with branch vessels. For more complex aneurysms exhibiting unfavourable anatomy, adjunct techniques, as outlined below, may be employed for treatment.

### 8.2. Balloon-Assisted Coiling (BAC)

In the treatment of complex, wide-neck aneurysms, it is difficult to achieve coil stability and dense packing with an unassisted coiling. Additional support is necessary in such situations to provide a scaffold. The technique of BAC was first described by Moret et al. in 1997 [42]. The procedure consists of the temporary inflation of a non-detachable balloon across the aneurysm neck during each coil placement (Figure 3A,B). It also stabilizes the microcatheter position during coiling and acts as a safety valve in the event of intra-operative rupture. Nowadays, a variety of balloons, including compliant, hypercompliant,

round-shaped, and double-lumen balloons, are available. The added advantage with the use of a double-lumen balloon is that it has separate inflation and working lumens and allows the placement of a stent at the end of procedure if required.

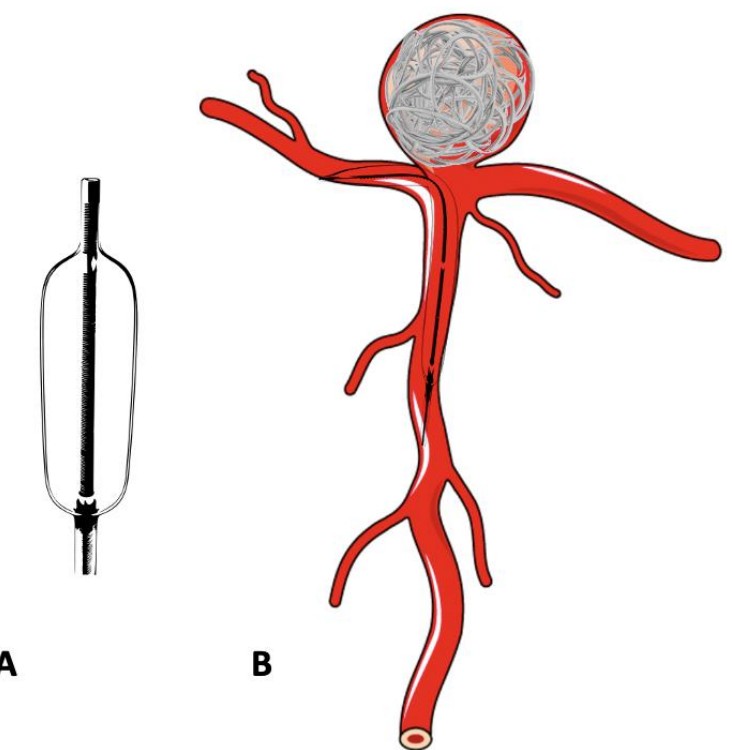

**Figure 3.** Schematic diagram of a double-lumen balloon and its use in the treatment of bifurcation aneurysm. (**A**) Double -lumen balloon. (**B**) Bifurcation aneurysm treated with balloon-assisted coiling.

### 8.3. Stent-Assisted Coiling

Though the balloon provides a necessary scaffold for coiling, it is temporary and removed at the end the end of the procedure. Hence BAC is inadequate in cases with extremely unfavourable aneurysmal anatomy. A stent which provides permanent support to coils overcomes this limitation at the cost of requiring long-term antiplatelet therapy to maintain the stent patency (Figures 4A and 5B). In 1997, Higashida et al. demonstrated the use of stents in surgically inoperable posterior circulation aneurysms [43]. The initially used stents were balloon mounted. The first self-expanding stent designed specifically for intracranial aneurysm treatment was the Neuroform stent (Stryker Neurovascular, Fremont, CA, USA) which received FDA approval in 2002 [44]. Stents which are nowadays available are high porosity, self-expanding stents with a highly flexible configuration to permit navigation across tortuous vessels whilst minimising vascular intimal injury. Furthermore, apart from their use in side wall aneurysms, single or multiple stents are used in different configurations, such as a X, Y, T, waffle-cone, or shelf, to provide the necessary scaffold at the neck of the aneurysm, which may not be possible with BAC alone [45,46].

The most troublesome complications with BAC and SAC are the risks of thromboembolic events and procedural aneurysmal rupture. The multicentre CLARITY trial compared the safety and efficacy of the remodelling technique (BAC and/or SAC) with conventional coiling in ruptured aneurysms. The study revealed better immediate postoperative occlusion rates in the remodelling group than conventional coiling (94.9% vs. 88.7%) with improved packing density (39.3% vs. 36.7%) despite unfavourable anatomy in the remodelling group. The equivalent rate of treatment-related complications (remodelling group 16.9% vs. conventional coiling 17.4%) and treatment-related cumulative morbidity and mortality (remodelling group 3.8% vs. conventional coiling 5.1%) was noted in both of the groups. A meta-analysis by Wang et al., comparing BAC and SAC, demonstrated no

difference between complete occlusion rates (COR) at the end of the procedure, rate of post-treatment complications, or rate of retreatment between groups. However, the COR at 6 months or later was better with SAC [47]. Piotin et al., found lower recurrence rates with SAC compared to conventional coiling [48]. The reason for improved outcomes with stent-assisted coiling may be explained by fluid dynamic studies, which revealed that stent placement causes a reduction of aneurysmal vortex speed and decreased interaction with parent vessel flow, depending on porosity [49]. This discovery subsequently led to the development of flow diverter stents.

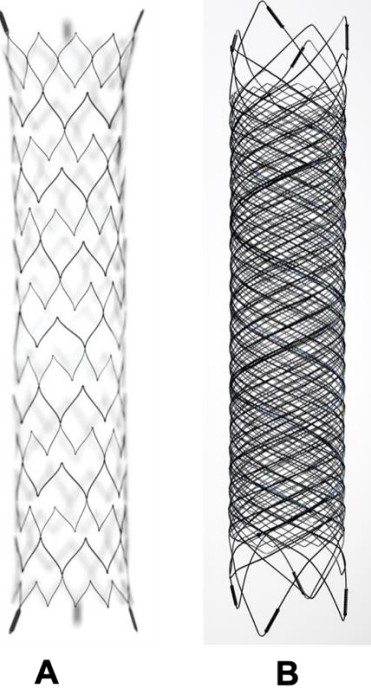

**Figure 4.** The difference between a conventional intracranial stent and flow diverter stent. (**A**) Conventional stent with higher porosity. (**B**) Flow diverter stent with reduced porosity.

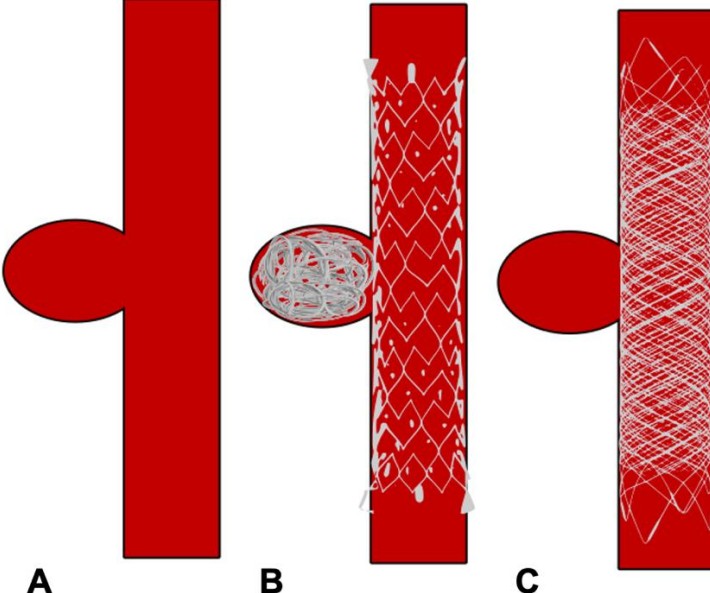

**Figure 5.** Schematic diagram showing treatment of side wall aneurysm. (**A**) Side wall aneurysm (**B**) Endovascular treatment using a conventional stent and coils (**C**) Endovascular treatment using flow diverter stent without coils.

### 8.4. Flow-Diverting Stents

Flow diverters (FD), working upon the principle of modifying intra-aneurysmal haemodynamic, are another substantial addition to the arsenal of aneurysm treatment. The design of flow-diverting stents is akin to conventional stents but with significantly lower porosity, serving two functions (Figure 4A,B). Firstly, it redirects blood flow away from the aneurysm into the parent vessel, promoting stasis and thereby natural thrombosis in the aneurysm (Figure 5). The thrombus acts like a scar which collapses over time. Secondly, FD provide a scaffold for endothelial growth, thereby isolating the aneurysm from the parent circulation [50]. By allowing neo-endothelisation to occur, FD heals the weakened abnormal arterial wall and therefore, provides a curative outcome as compared to other therapeutic options. Unlike surgical clipping and coil embolization which provide immediate protection against the future risk of rupture, the process of stasis, sac thrombosis and vessel healing by FD occurs slowly and can take up to 6 to 12 months after the treatment. However, compared to coiling, which has an overall recanalization rate of 20% and retreatment rate of up to 10%, very high (>90%) occlusion rates have been reported with FD on long-term follow-up [51–53].

The first FD introduced in the field of neurovascular intervention was the Pipeline Embolization Device (Medtronic Neurovascular, Irvine, CA, USA) in 2007, which received FDA approval in 2011. After its success and acceptance, a plethora of new devices have entered into the ever-growing field of FD. All of these devices work on the same haemodynamic principle, with some differences, mainly in terms of the number of wires, delivery system, and stent material used. Initial studies with FD focused on lesions such as giant, fusiform, difficult-to-treat aneurysms [54,55]. However, with increasing experience, there is an expanding list of indications for FD including complex wide-neck aneurysms, recanalized aneurysms, distal anterior circulation, and very small aneurysms which are otherwise challenging to treat by conventional endovascular options [56–58]. Though its use is precluded in acutely ruptured aneurysms due to the lack of immediate occlusion and need of antiplatelet therapy, a recently published meta-analysis (20 studies with 126 patients) on FD use in ruptured IA demonstrated good clinical outcomes (81%) with a complete occlusion rate of 90%. The majority (73%, 92/126) of ruptured aneurysms were treated with FD placement only [59]. Recently, the use of FD has been studied in middle cerebral artery (MCA) bifurcation aneurysms, with 68% occlusion at 6 months and 95% occlusion at 12 months along with 8.6% morbidity without any mortality [60]. So far there is insufficient evidence at present to support the routine use of FD in ruptured or in bifurcation aneurysms.

Complications associated with FD include thromboembolic events, bleeding risk associated with antiplatelet use, the risk of stent stenosis or thrombosis, and the risk of perforator occlusion inherent to the mechanism of action of FD. The International Retrospective Study of the Pipeline Embolization Device (IntrePED), in which 793 patients with 906 aneurysms were included, reported a 4.7% risk of ischemic stroke due to thromboembolic complications and a total of 8.4% for neurological morbidity and mortality [61]. Due to its endoluminal approach without accessing the aneurysm sac, FD placement reduces the possibility of intraprocedural aneurysm rupture inherent with conventional endosaccular coiling. However, they are associated with unique complications including distal intraparenchymal haemorrhage and delayed aneurysm rupture [62]. Though the exact aetiology for both of these complications is unknown, the postulated theory for intraparenchymal haemorrhage is that it may be a result of hemodynamic alterations secondary to device placement or the haemorrhagic conversion of microinfarcts. Delayed aneurysm rupture is probably due to inflammation related to intra-aneurysmal thrombus formation and/or hemodynamic changes in an unstable aneurysm induced by the placement of the device. In the IntrePED study, the rates of distal intraparenchymal haemorrhage and spontaneous rupture were about 2% and 0.5%, respectively. In the study, the lowest complication rates were seen when it was used to treat small ICA aneurysms, whilst the highest complications rates were noted with posterior circulation and giant aneurysms [61]. Hoe et al. reported the highest rate of delayed aneurysm rupture in giant, symptomatic aneurysms with a high

aspect ratio, associated with morbidity and mortality rates of >70% [63]. Nevertheless, FD are a viable and effective tool for the treatment of complex aneurysms with high occlusion rates and a good safety profile.

## 9. Newer Devices

Due to the limited utility of FD in acutely ruptured aneurysms and/or bifurcation aneurysms, the endovascular treatment of such aneurysms remains a challenge. The conventional approach consists of BAC or SAC. These conventional but technically complex methods are associated with a higher rate of incomplete occlusion, recanalization and retreatment, increased risk of intra-operative and post-operative complications, and risk of bleeding due to antiplatelet use and demand better operator skills [53,64]. Hence, nowadays, a major area of research focus is on the treatment of complex bifurcation aneurysms. The list of such devices is extensive, but notable ones include the intrasaccular flow disruption devices (WEB, MicroVention, Aliso Viejo, CA, USA), pCONus (Phenox, Bochum, Germany), PulseRider (Pulsar Vascular, San Jose, CA, USA), Endovascular Clip System (eClIPs, Evasc Medical systems, Vancouver, BC, USA), and Barrel vascular reconstruction devices (VRD, Medtronic). In addition to this, several innovative devices currently in different stages of development and trials are on the horizon.

### 9.1. Intrasaccular Flow Disruption Devices

The mechanism of intrasaccular flow disruption devices is similar to the intraluminal FD technology. When deployed inside the aneurysm they create flow stasis and thrombosis. The consequent advantage of intrasaccular location is the elimination of the need for dual antiplatelet therapy, which makes it suitable for use in acute ruptured aneurysm and reduces the risk of perforator or side branch occlusion. Woven EndoBridge (WEB, Sequent Medical, Palo Alto, CA, USA) and Medina Embolic Device (MED, Medtronic) are the two main devices known to be working on the principle of intrasaccular flow disruption, but they completely differ in terms of their design.

The WEB device has been available for clinical use in Europe and South America since 2010 and has undergone multiple improvements since then. The WEB device received US FDA approval in December 2018 [65]. WEB was initially introduced as a self-expanding, oblate-shaped, dual-layered (WEB DL), braided mesh of nitinol. It has now been redesigned into a single layer version (WEB SL) with a higher number of nitinol wires providing similar flow disruption effects (Figure 6A,B). Additional modifications include spherical (WEB SLS), barrel shapes (WEB SL), and enhanced visualisation (WEB EV) variants by using platinum-cored nitinol wires. It is fully retrievable and electrothermally detachable. Also, whilst initially designed for wide-neck bifurcation aneurysms, with recent technological advancement, it can now be used for side wall and small aneurysms [66]. The WEB-IT study in the USA reported 53.8% complete aneurysm occlusion and 84.6% adequate occlusion at 12 months for the treatment of 143 patients with wide-neck bifurcation aneurysms [67]. In a systematic review of WEB use in ruptured aneurysms, Brinkji et al. reported 87% adequate occlusion with 5.1% of patients needing retreatment. The study showed a low rate of rebleeding and clinical complications [68].

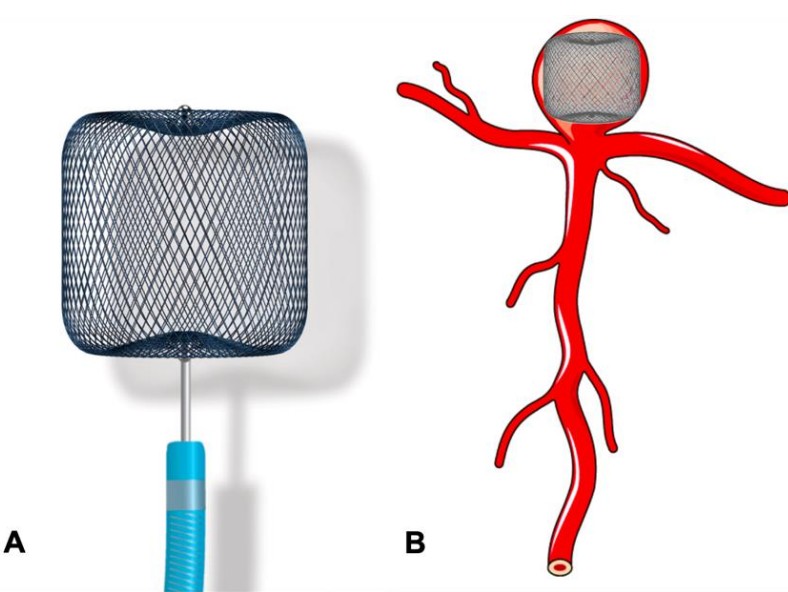

**Figure 6.** Use of the WEB device in the treatment of bifurcation aneurysm. (**A**) Woven EndoBridge (WEB) device. (**B**) Bifurcation aneurysm treated with WEB device.

Another device, the Contour Neurovascular System (Cerus Endovascular, Fremont, CA, USA) has recently received CE mark approval for the treatment of IA. It is a fine mesh braided design, which is deployed with its base at the neck of the aneurysm and body within the aneurysm. It provides a combination of flow diversion and flow disruption through a single device implant. The recent meta-analysis of early experience with the Contour device showed a short procedure time with adequate occlusion rates and functional independence [69]. Another device working on the principal of flow disruption (Artisse-Luna, Medtronic, Irvine, CA, USA) is currently under clinical evaluation and preliminary results are encouraging [70,71]. High rates of neck remnant, difficulty in treatment of lobulated aneurysms, limited clinical data, and lack of device availability in the international market are current limitations to the widespread use of some of the intrasaccular devices. However, the technology of intrasaccular flow disruption appears promising, particularly in view of the lack of need for long-term antiplatelet therapy and thereby feasibility for use in acute ruptures.

### 9.2. pCONus

The pCONus devices represent the evolution of the previously described waffle-cone technique [72]. It is another extensively used newer device and consists of a self-expanding laser cut stent, with a distal crown of four petals deployed in the aneurysm and its base with six polyamide fibres at the level of the neck [73]. It permits stable coil placement by providing a mechanical barrier at the level of the aneurysm neck. A recently introduced 2nd generation pCONus device has six distal petals without polyamide fibres (Figure 7). The redesigned crown in pCONus 2 allows it to accommodate steep angles between the parent vessel and aneurysm sac, whilst also providing greater metal coverage inside the sac to aid aneurysm coiling. The unique design of pCONus allows it to have <5% metal coverage inside the parent artery. A study by Fischer et al. (25 aneurysms; 18 unruptured; 7 ruptured) showed successful deployment in 96% of cases with adequate occlusion (total occlusion and neck remnant) in 81% of cases at follow-up and a procedure-related permanent morbidity for 4% of patients [74].

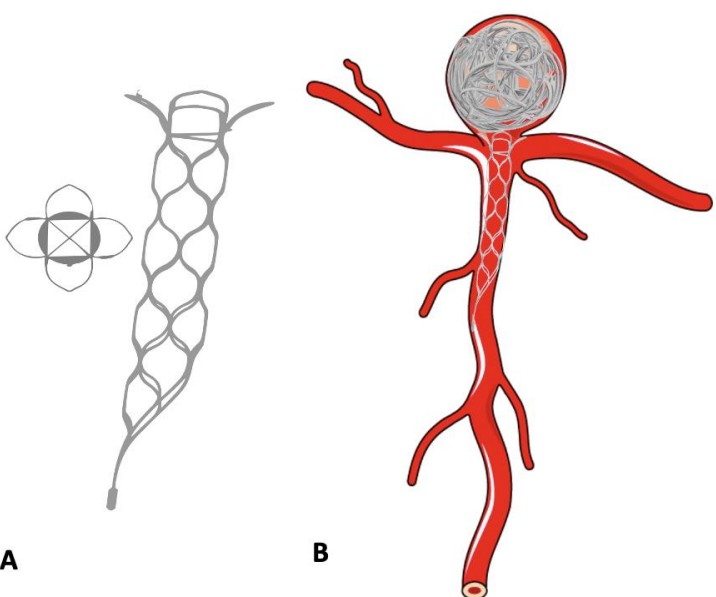

**Figure 7.** Use of the pCONus device in the treatment of bifurcation aneurysm. (**A**) pCONus device. (**B**) Bifurcation aneurysm treated with pCONus device and detachable coils.

*9.3. PulseRider*

PulseRider is another device working on the principle of the waffle-cone technique. It is a self-expanding, fully retrievable, nitinol stent which anchors in the parent vessel and bridges the neck when deployed (Figure 8). The device is available in T and Y configurations and has a metal load of 5–7%. The recently published ANSWER trial showed successful deployment in all patients with an immediate RRC score of I or II in >80% patients without any clinically significant complication [75].

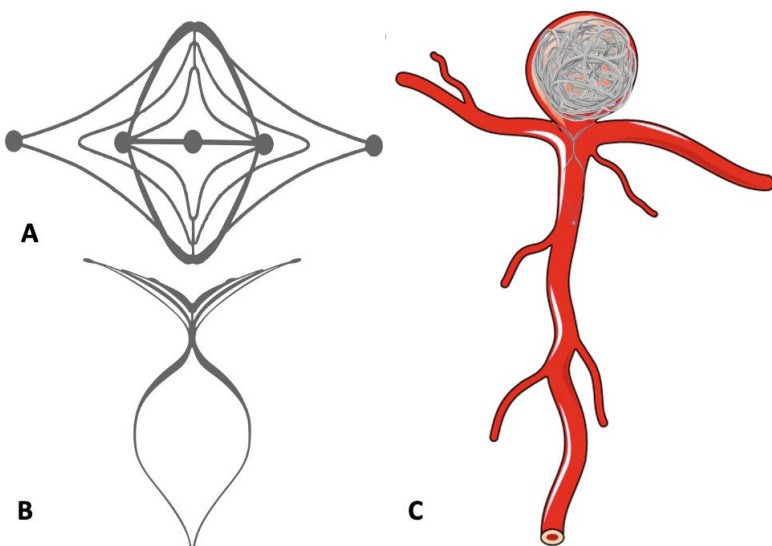

**Figure 8.** Use of the Pulse Rider device in the treatment of bifurcation aneurysm. (**A**,**B**) Pulse Rider Device. (**C**) Bifurcation aneurysm treated with Pulse Rider Device and detachable coils.

*9.4. eCLIpse*

The endovascular clip system, eCLIPs device, is a laser-cut, non-circumferential device, that is comprised of an 'anchor' to conform to the neck and a 'leaf segment' with moveable ribs designed to allow delivery through a coiling microcatheter (Figure 9). The leaf segment has a metal coverage in the range of 23–42% over the neck and acts as a flow disruptor by reducing the water hammer effect. It also provides a scaffold for endothelial growth.

Though the first generation eCLIPs device was balloon mounted, the redesigned 2nd generation eCLIPs device is self-expanding, microcatheter-deliverable, fully retrievable and self-orientating. The data published by Marotta et al. with eCLIPs is encouraging [73,76].

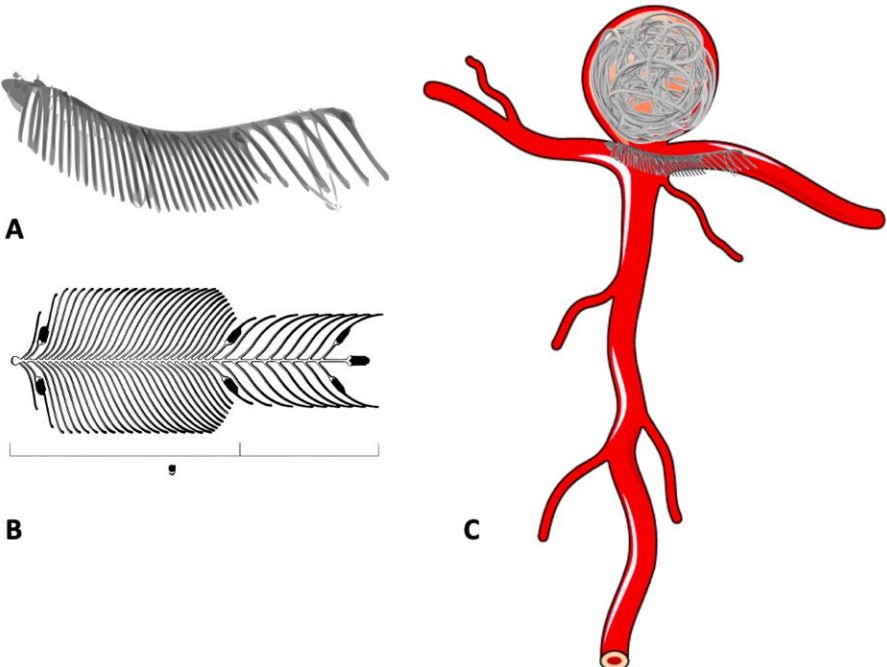

**Figure 9.** Use of the eCLIPs device in the treatment of bifurcation aneurysm. (**A,B**) eCLIPs device. (**C**) Bifurcation aneurysm treated with eCLIPs device and detachable coils.

### 9.5. Barrel Stent

The Barrel vascular reconstruction device is a laser cut, self-expanding, resheathable stent with a pre-existing enlarged barrel-like mid-section. This structure can help to provide better neck coverage compared to a single conventional stent, narrows the neck by bracing the walls of the parent vessels, and allows easier microcatheter navigation [70,73]. A recently published multicentric study showed successful deployment of the device in all cases with 90% immediate and 95% midterm follow-up occlusion rates [77].

Attempts are also being made to use liquid embolic agents, which are commonly used in treating cerebrovascular malformations, to treat IA [78]. Liquid embolic agents, form a spongy solid cast within minutes of contact with blood. Therefore, these agents, when injected, first conform to the shape of the aneurysm and then solidify [79]. The most important risk with the procedure is the liquid embolic refluxing into the parent artery. To date, there is a paucity of data available regarding the use of liquid embolics in aneurysm treatment. The use of covered stents for treating aneurysms has also been demonstrated in a few clinical studies [80]. The main advantage of covered stents over FD is the immediate protection against rupture. They also avoid the mass effect which accompanies conventional coiling. Disadvantages include limited flexibility and deliverability of covered stents, endoleak risk, high restenosis, and closure of side branches or perforating arteries originating from the covered arterial segment. Apart from newer devices, improvement in catheters (guiding, intermediate, and microcatheters) as well as guidewire technology with improved trackability and better proximal and distal support, and the availability of newer antiplatelet drugs have improved outcomes following endovascular treatment.

In the coming years, artificial intelligence (AI) stands at the forefront of revolutionizing intracranial aneurysm evaluation and treatment. In medical imaging, AI algorithms can enhance the accuracy of aneurysm detection and characterization, aiding clinicians in diagnosis and treatment planning. As artificial intelligence (AI) continues to advance, it holds the potential to aid in assessing rupture risk, prioritizing clinical therapy strategies, and pre-

dicting treatment outcomes for intracranial aneurysms. Although we have not yet reached the threshold for routine clinical application, with the availability of larger datasets and ongoing research, AI holds potential to contribute to patient-centric intracranial aneurysm management in the future [81].

## 10. Current Evidence for the Selection of Appropriate Treatment Strategy

Over the years, the evolution of surgical and endovascular therapy, improvement in the techniques of neuro-anaesthesia, aggressive treatment of vasospasm, widespread availability of neurocritical care units, and advanced neuroimaging has improved the outcomes of aneurysm treatment. Although the emergence of endovascular therapy in 1990 has reduced the role of open surgery over last few years, the high chances of recanalization, high cost of the devices, and need for long-term follow-up are some of the pitfalls of interventional therapy. Hence, for certain groups of aneurysms, surgical therapy is still a cost-effective option with better occlusion rates. The development of newer devices, including flow diverters, has started a new era in the field of endovascular therapeutics, which has allowed the treatment of previously untreatable or difficult-to-treat aneurysms with a good success rate and less complications.

With the availability of different treatment modalities, each with their own advantages and drawbacks, the selection of appropriate strategy depends on multiple factors, which include not only patient and aneurysm-related factors, but also the availability of hospital resources including the skill and experience of the operating surgeons. Hence, potential risks and benefits of a given approach must be individualized to the patient and their lesions. Amongst the patient-related factors, age is one of the important determinants of the modality of management. Although in general old patients have poorer outcomes independent of the modality of treatment, it is evident that endovascular treatment is better tolerated in this group of patients [82]. However, sometimes, the presence of tortuous arteries may make endovascular management difficult in an older patient. Lower rates of epilepsy, infections, and pulmonary complications have been noted in patients > 65 years of age who underwent coiling when compared to clipping [82]. Also, the presence of comorbidities, poor clinical condition, and prior use of antiplatelets or anticoagulants favour endovascular treatment [83]. The aneurysm-related factors governing treatment modality are aneurysm size, neck size, morphology of aneurysm sac, relation to branch vessel, and location of the aneurysm. Traditionally, aneurysms with a large size, wide neck (>4 mm), dome-to-neck ratio < 1.5, lesions incorporating branch vessels, and bifurcation location were considered as poor candidates for endovascular treatment due to high rates of incomplete occlusion and associated complications. However, with availability of newer intracranial stents, flow diverters, intrasaccular flow disruptors, and various other bifurcation devices, more and more of these aneurysms are now being treated with an endovascular modality. Posterior circulation aneurysms are traditionally considered as difficult to access surgically and associated with a high morbidity and mortality, hence endovascular treatment is the preferred modality in this group. Another aneurysm type where endovascular therapy has completely overshadowed surgical treatment is the dissecting aneurysm, in which overlapping stents or flow diverters are now the preferred modality to trap the dissecting flap. On the other hand, a surgical modality is favoured, particularly in young adults, with ruptured anterior circulation bifurcation aneurysms due to ease of access and better long-term occlusion rates [84]. In situations where SAH is associated with large intraparenchymal or intradural/extradural bleeding, hematoma evacuation can be combined with definitive surgical treatment when feasible. Another unrelated but important factor which also guides the choice of treatment modality is the economic impact of the procedure, particularly in developing countries. A study by Lad et al. looking at the financial aspect of aneurysm treatment suggested that, although surgical treatment is associated with a higher initial cost, this difference wanes over the next 2 to 5 years due to a higher number of follow-up angiograms and outpatient costs in endovascular groups [85].

Prior to 1990, ruptured aneurysms were almost exclusively treated by surgical clipping. The advent of GDC coil provided an alternative option for aneurysm treatment. The International Subarachnoid Aneurysm Trial (ISAT) was the first randomised, multicentre, prospective trial, which assessed the safety and efficacy of endovascular coiling with standard neurosurgical clipping for aneurysms judged to be suitable for both treatments [86]. Survival free of disability at 1 year was significantly better in patients who underwent endovascular coiling (disability rate 25% with coiling vs. 36% with surgery). In 2009, the assessment of the long-term risks of death, disability, and rebleeding found a significantly lower risk of death at 5 years (11% vs. 14%), and small but increased risk of rebleeding in patients who underwent coiling (ten in the coiling group and three in the clipping group) [87]. Again, in 2015, 18-year follow-up data from the same study group showed that the probability of disability-free survival was significantly greater in the endovascular group than in the neurosurgical group at the cost of a small increase in the risk of bleeding [88]. The authors suggested that surgical clipping might be favoured over coiling in younger patients in view of better occlusion rates. However, generalisation of this data to the entire SAH population is not possible because of its selection bias, with 77.6% of aneurysms excluded from the study due to failure to meet inclusion criteria. Additionally, the majority of patients randomised had anterior circulation aneurysms (97.3%), small aneurysms (90% < 10 mm), and were in good clinical condition prior to treatment.

To overcome the selection bias in the ISAT, the retrospective, randomised, controlled trial Barrow Ruptured Aneurysm Trial (BRAT) was designed to reflect the real-world practicalities of ruptured aneurysm treatment in North America [89]. One year after treatment, coil embolization resulted in fewer poor outcomes than clip occlusion (23.2% vs. 33.7%) in the prospectively enrolled study population of 500 patients. However, at 3 years, the difference in risk of a poor outcome had decreased from that observed at 1 year and was no longer significant (30% vs. 35.8%). Another meta-analysis performed by Hui Li et al. concluded that coiling reduced the 1-year unfavourable outcome rate, particularly in patients with a good preoperative grade compared to those with a poor preoperative grade. The risk of vasospasm was more common with clipping, while coiling led to a greater risk of rebleeding [90].

Similar results have been obtained with studies which focussed exclusively on the unruptured aneurysm group. A study by McDonald et al. analysed 4899 unruptured aneurysm patients (1388 clipping, 3551 coiling) treated at 120 hospitals in the USA, between 2006 and 2011. The study found that patients who underwent clipping had a higher likelihood of unfavourable outcomes, including ischemic complications, haemorrhagic complications, discharge to long-term care, postoperative neurological complications, and ventriculostomy, when compared to coiling, with a similar likelihood of in-hospital mortality [91]. An extensive meta-analysis which included studies with both ruptured (117,495 individuals including 2918 patients from RCTs, 11,303 patients from observational studies, and 103,274 patients from database registry) and unruptured aneurysm (108,277 patients including 7487 from observational studies and 100,790 from database registry studies) revealed higher independent outcome and lower mortality after coiling compared to clipping [92].

In conclusion, the evolution of surgical and endovascular therapies, along with advancements in clinical care, has significantly enhanced outcomes in aneurysm treatment. Given this plethora of treatment modalities now available, each with their own advantages and drawbacks, the selection of an appropriate strategy has become inherently complex. The advent of endovascular therapy in the 1990s has notably diminished the prominence of open surgery. Nevertheless, coil embolization has established itself as a mainstay modality of aneurysm management, which is likely to be amplified further with the introduction of innovative endovascular devices, including flow diverters and intrasaccular flow disruptors. However, interventional therapy is not without its pitfalls, including the risks of recanalization, high procedural and equipment costs, and the necessity for long-term follow-up. Consequently, for specific groups of aneurysms, surgical therapy remains a

cost-effective option with superior occlusion rates. The development of minicraniotomy techniques, endoscope-assisted microsurgeries, and intraoperative angiography has introduced new dimensions to aneurysm clip surgery. While the long-term results of emerging endovascular devices and surgical techniques are pending, it is crucial to observe how these advancements perform over time. We strongly believe that team-based, individualized treatment after thorough discussion with patients and their families regarding the different options and expected outcomes should be the guiding principle for the successful aneurysm treatment.

**Author Contributions:** Conceptualization, A.S.D. and C.H.; methodology, C.H.; writing—original draft preparation, A.S.D., G.S. and S.M.P.; writing—review and editing, S.M.P. and A.C.A.; visualization, A.S. and A.H.K.; supervision, C.H. All authors have read and agreed to the published version of the manuscript.

**Funding:** This research received no external funding.

**Institutional Review Board Statement:** This study did not require ethical approval.

**Informed Consent Statement:** Not applicable.

**Data Availability Statement:** No new data were created or analyzed in this study. Data sharing is not applicable to this article.

**Conflicts of Interest:** The authors declare no conflicts of interest.

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
