# Peer review of "The Management of Intracranial Aneurysms: Current Trends and Future Directions"

_2035-8377, doi:10.3390/neurolint16010005_

Round 1

Reviewer 1 Report

Comments and Suggestions for Authors

This depends on the journal, but using a Gmail email address may not be considered professional.

The provided graphic in Fig. 1 contains valuable and interesting information. However, it could benefit from a more polished and professional appearance. Please find an improved way to enhance the visual presentation of the graphics. And if you have relied solely on one source for information to create this graphics, please remember to include a citation.

Speaking of Figures, to enhance the comprehensibility of your review paper, incorporate more visuals, especially in the introduction section. By utilizing graphics, you can effectively introduce readers unfamiliar with the topic of your manuscript to its main subject and associated issues. It is important to maintain a professional approach while creating these visuals so that they serve as helpful tools for learning about the topic at hand.

Discuss how the presence of additional medical conditions (comorbidities) can create challenges when it comes to treating Intracranial Aneurysms, e.g.: 10.3390/applmech4020033

Examine and discuss the development phases of the techniques and devices mentioned. How are they evaluated before being incorporated into clinical practice? Is mechanical testing conducted? Are computational simulations employed in the evaluation process?

Explore and discuss the potential application of artificial intelligence in supporting the advancement and evaluation phases of the described methods and devices.

Author Response

Dear reviewer, 

Thank you for reviewing our manuscript titled Management of Intracranial Aneurysms: Current Trends and Future Directions. We appreciate your recommendations. As advised, corrections have been made and we are now submitting manuscript.

  1. This depends on the journal, but using a Gmail email address may not be considered professional.

We understand your concerns and will add the institutional email id. 

2. The provided graphic in Fig. 1 contains valuable and interesting information. However, it could benefit from a more polished and professional appearance. Please find an improved way to enhance the visual presentation of the graphics. And if you have relied solely on one source for information to create this graphics, please remember to include a citation.

The graphic has been created based on timelines in evolution of aneurysm treatment and created by one of the co-authors. The necessary data has been collected from multiple sources.

3. Speaking of Figures, to enhance the comprehensibility of your review paper, incorporate more visuals, especially in the introduction section. By utilizing graphics, you can effectively introduce readers unfamiliar with the topic of your manuscript to its main subject and associated issues. It is important to maintain a professional approach while creating these visuals so that they serve as helpful tools for learning about the topic at hand.

We welcome your suggestions about the figures. Multiple Schematic Figures have been added to improve the comprehensibility of the paper.

4. Discuss how the presence of additional medical conditions (comorbidities) can create challenges when it comes to treating Intracranial Aneurysms, e.g.: 10.3390/applmech4020033

We welcome this suggestions. Additional text has been added about patient related factors in the aneurysm treatment and we have referenced the article you have suggested. 

5. Examine and discuss the development phases of the techniques and devices mentioned. How are they evaluated before being incorporated into clinical practice? Is mechanical testing conducted? Are computational simulations employed in the evaluation process?

We appreciate your insightful suggestions regarding the examination and discussion of the development phases, evaluation processes, and the potential incorporation of techniques and devices into clinical practice. However, after careful consideration, we believe that delving into the specific development phases, mechanical testing, and computational simulations falls outside the intended scope of the current discussion.

6. Explore and discuss the potential application of artificial intelligence in supporting the advancement and evaluation phases of the described methods and devices

We have added some additional text discussing role of AI in aneurysm treatment as you have suggested. 

Reviewer 2 Report

Comments and Suggestions for Authors

This is a review paper on the management of intracranial aneurysms. The paper touches only briefly on epidemiology and mechanisms, before launching into in-depth discussion on occlusive management of saccular aneurysms.

Ther are some issues the authors need to attend to:

Title, lines 95-96 (major) – I see only saccular aneurysms are addressed - should the title be changed?

Line 46 - what is ‘cross-sectional neuroimaging’?

Lines 73-79 – a table with this info would be helpful to readers

Line 83 – what is ‘rupture status’? Ruptured and unruptured?

Lines 82-87 – a table with this info would be helpful to readers

Line 170 – add ‘Arterial occlusion and…’

Fig 1 – 1994 – Fischer - the box is too small for the text?; 2003 - A Rabbe etc  to be in bold?

Major - The authors need to present data from blinded randomised controlled trials to show efficacy and safety

Major – the authors should provide treatment algorithms

Major – surprisingly, little on complications of these techniques

Major – regrettably, no discussion that with more interventional work, the neurosurgeons will become less skilled with time and will be incompetent in managing the complex aneurysm that endovascular techniques can’t help

Major - I sadly see no discussion on the general management of the patient eg blood pressure vasospasm etc – the authors are only interested in occluding the aneurysm – the tile should be revised? ‘occlusive techniques for saccular intracranial aneurysm management’

Grammar – line 47 - ‘depends’; line 51 - no need for ‘to’; line 55 – ‘in the initial’; line 57 ‘within the first’; line 63 - ‘provide a comprehensive’ – this is only from the Introduction, these errors continue throughout the text

Comments on the Quality of English Language

Needs attention

Author Response

Dear reviewer, 

Thank you for reviewing our manuscript titled Management of Intracranial Aneurysms: Current Trends and Future Directions. We appreciate your recommendations. As advised, corrections have been made and we are now resubmitting manuscript.

1. Title, lines 95-96 (major) – I see only saccular aneurysms are addressed - should the title be changed?

We did focus mainly on saccular aneurysm consider it being the largest group. But throughout the text we did discuss bypass surgery, wrapping technique and use of flow diverters which are commonly used treatment modalities for the dissecting and fusiform aneurysms as well. However, if you insists we are happy to change the title accordingly. 

2. Line 46 - what is ‘cross-sectional neuroimaging’?

Typically, MRI and CT are considered cross sectional imaging as they produce an image in the form of a plane through the body with the structures cut across. (In comparison to fluoroscopy and X-rays which does not.)

3. Lines 73-79 – a table with this info would be helpful to readers

We welcome you comment and table has been added. 

4. Line 83 – what is ‘rupture status’? Ruptured and unruptured?

Corrected in the main text.

5. Lines 82-87 – a table with this info would be helpful to readers

We welcome you comment and table has been added. 

6. Line 170 – add ‘Arterial occlusion and…’

Added

7. Fig 1 – 1994 – Fischer - the box is too small for the text?; 2003 - A Rabbe etc  to be in bold?

Corrected

8. Major - The authors need to present data from blinded randomised controlled trials to show efficacy and safety

Major – the authors should provide treatment algorithms

Major – surprisingly, little on complications of these techniques

Major – regrettably, no discussion that with more interventional work, the neurosurgeons will become less skilled with time and will be incompetent in managing the complex aneurysm that endovascular techniques can’t help

Major - I sadly see no discussion on the general management of the patient eg blood pressure vasospasm etc – the authors are only interested in occluding the aneurysm – the tile should be revised? ‘occlusive techniques for saccular intracranial aneurysm management’

We appreciate recommendations. We have added data comparing major randomized trials. Some data about complications has been added to individual treatment modalities.

Regarding the suggestion to broaden the scope to include general patient management aspects of SAH as well as potential impact of increased interventional work on the skillset of neurosurgeons are intriguing points. We understand the importance of a holistic approach to patient care. Unfortunately, due to word count limitations and the specific focus of our article on occlusive techniques, we had to make choices about content inclusion. We appreciate your suggestion to revise the title to reflect a more specific focus, and we will certainly consider it in our future works.

9. Grammar – line 47 - ‘depends’; line 51 - no need for ‘to’; line 55 – ‘in the initial’; line 57 ‘within the first’; line 63 - ‘provide a comprehensive’ – this is only from the Introduction, these errors continue throughout the text

Corrected

Reviewer 3 Report

Comments and Suggestions for Authors

Dear Authors,

thank you for this nice and comprehensive review.

To my opinion they are no issues to discuss, however, if you agree, I would just add a data about the rate of thromboembolic complications when using FD, as this might be one of the most feared complication of this technique.

Author Response

Dear reviewer, 

Thank you for reviewing our manuscript titled, Management of Intracranial Aneurysms: Current Trends and Future Directions. 

We appreciate your kind words and recommendations. As advised, corrections have been made with addition of rates of thromboembolic complications noted in IntrePED study. We are now resubmitting manuscript. Thank you. 

Round 2

Reviewer 1 Report

Comments and Suggestions for Authors

I appreciate your attention to most of my comments. This reviewer does not have any further feedback to provide.

Reviewer 2 Report

Comments and Suggestions for Authors

Please provide a 'track changes' version so that the amendments can be easily seen

Comments on the Quality of English Language

ok

Round 3

Reviewer 2 Report

Comments and Suggestions for Authors I apologise for not reviewing the paper in time as i was extremely busy The authors have addressed my queries Thank you